# Development of Smart Weighing Lysimeter for Measuring Evapotranspiration and Developing Crop Coefficient for Greenhouse Chrysanthemum

**DOI:** 10.3390/s22166239

**Published:** 2022-08-19

**Authors:** Atish Sagar, Murtaza Hasan, Dhirendra Kumar Singh, Nadhir Al-Ansari, Debashis Chakraborty, Mam Chand Singh, Mir Asif Iquebal, Amit Kumar, Pankaj Malkani, Dinesh Kumar Vishwakarma, Ahmed Elbeltagi

**Affiliations:** 1Division of Agricultural Engineering, ICAR—Indian Agriculture Research Institute, New Delhi 110012, India; 2Centre for Protected Cultivation Technology, ICAR—Indian Agriculture Research Institute, New Delhi 110012, India; 3Department of Civil, Environmental and Natural Resources Engineering, Lulea University of Technology, 97187 Lulea, Sweden; 4Division of Agricultural Physics, ICAR—Indian Agriculture Research Institute, New Delhi 110012, India; 5Indian Agricultural Statistics Research Institute, Indian Council of Agricultural Research, New Delhi 110012, India; 6Department of Irrigation and Drainage Engineering, G. B. Pant, University of Agriculture and Technology, Pantnagar 263145, India; 7Agricultural Engineering Department, Faculty of Agriculture, Mansoura University, Mansoura 35516, Egypt

**Keywords:** weighing lysimeter, evapotranspiration, crop coefficient, shallow rooted crop, greenhouse chrysanthemum

## Abstract

The management of water resources is a priority problem in agriculture, especially in areas with a limited water supply. The determination of crop water requirements and crop coefficient (K_c_) of agricultural crops helps to create an appropriate irrigation schedule for the effective management of irrigation water. A portable smart weighing lysimeter (1000 × 1000 mm and 600 mm depth) was developed at CPCT, IARI, New Delhi for real-time measurement of Crop Coefficient (K_c_) and water requirement of chrysanthemum crop and bulk data storage. The paper discusses the assembly, structural and operational design of the portable smart weighting lysimeter. The performance characteristics of the developed lysimeter were evaluated under different load conditions. The K_c_ values of the chrysanthemum crop obtained from the lysimeter installed inside the greenhouse were K_c_ ini. 0.43 and 0.38, K_c_ mid-1.27 and 1.25, and K_c_ end-0.67 and 0.59 for the years 2019–2020 and 2020–2021, respectively, which apprehensively corroborated with the FAO 56 paper for determination of crop coefficient. The K_c_ values decreased progressively at the late-season stage because of the maturity and aging of the leaves. The lysimeter’s edge temperature was somewhat higher, whereas the center temperature closely matched the field temperature. The temperature difference between the center and the edge increased as the ambient temperature rose. The developed smart lysimeter system has unique applications due to its real-time measurement, portable attribute, and ability to produce accurate results for determining crop water use and crop coefficient for greenhouse chrysanthemum crops.

## 1. Introduction

The actual determination of both reference evapotranspiration (ET_o_) and crop evapotranspiration (ET_c_) is necessary for estimating crop irrigation water demands, irrigation scheduling, vegetation monitoring, hydrological studies and irrigation system designs [1]. FAO irrigation and drainage papers 24 and 56 provide guidelines for crop coefficients and water-use functions for common crops grown around the world [2,3]. The varietal, cultural and environmental conditions might fluctuate significantly among locations; thus, the function generated for one environment would not be suitable for another location [3].

Measurement of ET_c_, i.e., actual crop irrigation water requirement throughout the growing season, is required for the determination of the crop coefficient. K_c_ values are then calculated as the ratio of crop evapotranspiration (ET_c_) to reference evapotranspiration (ET_o_) and thus a relationship is established to generate K_c_ values for the entire crop growing season.

Weighing lysimeters are extensively used to estimate and study crop water use patterns throughout crop growth and thereby standardize reference evapotranspiration models for the localized area to estimate crop-coefficient data for specific crops [4]. The amount of water that cultivated areas in a field require for evapotranspiration might be precisely measured using a weighing lysimeter. The applicability of lysimeter for estimating the crop coefficient of various crops has been widely reported worldwide, including corn crop in Spain [5], pulse in India [6], rice and sunflower in India [7], cotton and wheat in the USA [8] and wheat in China [9].

Although indirect micrometeorological methods, such as the Bowen ratio technique and Eddy covariance, are gaining popularity for expedite measurement of crop evapotranspiration, direct measurement of ET_c_ through lysimeters (weighing or drainage) is still considered a valid standard method [3,10,11,12,13,14,15]. Furthermore, based on the existing resources, uses and experimental necessity, different types and sizes of lysimeters are designed to investigate and study the water and solute transport in the vadose zone and estimate the ET_c_ rates [16,17]. The lysimeters are of different shapes, including rectangular shape [18,19,20,21], circular shape [22], and square shape [23,24]. In addition, investigators have used a small area (0.006 m^2^) of lysimeter, known as a micro lysimeter, to estimate evaporation from the soil surface, as well as evapotranspiration [25]. Lysimeters also find applications in monitoring nutrient movement through the soil profile, accessing the leached water quality below the root zone.

Lysimeter effectiveness in estimating ET_c_ from a cultivated area depends on accurate lysimeter operation, management and installation [26,27]. According to the needs of the experiment, various researchers have discussed and explained different lysimeter types [24]. Weighing lysimeters generates data based on variation in mean soil moisture, which is estimated by the variation in the change in weight of the lysimeter before and after a specified period [28]. The combination of soil moisture variation with other soil water balance components, such as precipitation, irrigation and drainage over a specified period of time, gives the mean crop ET_c_ rate [29,30].

The earlier classification of lysimeters included intermittent weighing and continuous weighing [31], afterwards they were together termed as weighable lysimeters. The key difference between them is the time lag between the two consecutive weight measurement readings. Continuous weighing lysimeters have high installation costs and require skilled manpower; thus, despite their precision and accuracy, their application is limited. The weighing mechanism for these lysimeters is capable of monitoring weight changes at short intervals; however, they require permanent fixation in the field and hence are non-portable. The time interval between two successive readings is generally 1 day or longer [32].

The primary objective of a lysimeter is to establish a favorable and controlled environment that is identical to field conditions for the measurement of water balance [33]. Regarding plant height, microenvironment, nutrient availability, soil moisture, root density, and other factors, the soil–plant system inside the lysimeter should be approximately similar to that of the surrounding area [32,34]. It is very difficult to exactly match the soil and water environment inside the lysimeter with respect to the field conditions. To mitigate this type of problem, care should be done at all steps, i.e., from designing the lysimeter to its construction, calibrations, fabrications, installation and management in the field. For any crops/plants grown in a container, the volume of available soil may be limited to a normal rooting profile. In addition, the availability of moisture at the bottom of the lysimeter is greater compared to the field at the same depth, unless an effective drainage system is provided to efficiently remove the extra water [35].

In the past, the weighing lysimeter extracted measurements with a relatively high degree of accuracy using electrical circuitry and mechanical mechanisms. This type of lysimeter required regular maintenance and care and had expensive initial and operational costs. With the advances and developments in data logging types of equipment, electronics science, and strain-gauge loadcell mechanisms, the design of a smart lysimeter is possible, which is relatively less expensive, reliable and accurate and requires minimum maintenance.

The main objectives of this research are to describe the design, construction and installation of a smart weighing lysimeter to measure the crop evapotranspiration (ET_c_) for shallow–rooted greenhouse crops. The performance of the developed lysimeter was evaluated for greenhouse chrysanthemum flower crops. This could also be used for modeling water and nutrient movement for shallow rooted crops.

## 2. Materials and Methods

### 2.1. Descriptions of the Study Area

The present study was carried out in a forced ventilated greenhouse at the Centre for Protected Cultivation Technology, ICAR-Indian Agricultural Research Institute, New Delhi (Figure 1) during the September to February months for two successive years (2019–2020 and 2020–2021). The study site, which was 229 m above mean sea level, was located between latitudes 28°37′ and 28°39′ N and longitudes 77°09′ and 77°11′ E. Experimental soil in the greenhouse was classified as sandy loam at varying soil depths of 0–30 and 30–60 cm. The average bulk density of the soil was 1.46 g/cm^3^. The average EC of the saturated extract and the pH of the soil of the experimental site were 0.29 dS/m and 7.5, respectively. The experimental site experiences cold winters and a semi-arid, sub-tropical environment with hot and dry summers. The hottest months were May and June, with maximum temperature ranges between 40 and 46.5 °C. The coldest month was January, with minimum temperatures between 3.5 and 6.8 °C. The average open pan evaporation reached as high as 13.8 mm/day during May. However, it was reduced to 1.1 mm/day during January. The description of planting and harvesting dates along with seasonal weather data for the year 2019–2021 are presented in Table 1 for greenhouse chrysanthemum.

### 2.2. Moisture Characteristics of Soil

The moisture characteristics of the soil in the experimental plot were determined with the help of Pressure Pate Apparatus to establish the water holding capacity and soil moisture at the field capacity level. Soil moisture at different suctions and at different depths of soil is presented in Figure 2. The field capacity and permanent wilting point of the soil were 22.4 and 8.53% (soil moisture content on a volumetric basis), respectively.

### 2.3. Components of the Weighing Lysimeter

The major components of the developed weighing lysimeter are as follows:A cultivation tank that contains soil from the same plot to replicate natural conditions, as well as to calculate the crop’s coefficient (K_c_) and evapotranspiration throughout different growth stages of the crop.A collection tank for collecting and measuring the water that has percolated through the media.Three wheels are assembled with three loadcells on which the lysimeter’s total load (soil + own weight) acts and determines the weight variation of the soil mass and the drainage tank.An Arduino display assembly with a data logger was used to obtain the measured data.

The following sections describe the design of the smart weighing lysimeter, including the installation process and structural analyses.

#### 2.3.1. Design and Construction of a Weighing Lysimeter

The Weighing lysimeter was designed with the aim of simple installation, minimum maintenance requirements and low constructional cost. It consisted of four major components: tank, loadcell assemblies, Arduino assembly and drain system. All the materials of the lysimeter elements were made of mild steel. The cultivation tank contained soil that was taken from the same field, and inside it, the plants were grown. The cultivation tank rests directly with the support of loadcells assembled on three wheels. The cultivation tank’s weight was sustained and monitored by the loadcell assemblies. The drainage mechanism allowed the removal of any extra water built up in the cultivation tank.

The report by Allen et al. [1] was used as the foundation for the design of the lysimeter, which was adequately adjusted to relocate the loadcells. The cultivation tank was mounted above ground and the loadcells were not directly exposed to the outside environment, resulting in a much smaller change in temperature, which did not affect the performance of loadcells and lysimeter measurements under greenhouse condition [36]. Minimization of the thermal/heat effects on loadcells by keeping the loadcells under the cultivation tank beneath the soil surface was included as integral design criteria. The lysimeter was designed for estimating or to generate the crop ET_c_, provided with a cultivation tank with a surface area of dimensions 1 m wide × 1 m long and a tank depth of 0.6 m. Figure 3 shows a drawing with top and side views. Mild steel sheets were used to construct the tank, and angle iron was used to hold each corner. A standard 6-mm (15/64-in) mild steel plate was used to construct the four side walls and bottom plate. Angle irons measuring 50 × 50 × 6 mm (2 in) were welded to the bottom and side plates to add strength and prevent the plates from bending. A cultivation tank was constructed by continuously welding the side and bottom plates together at each corner, creating a watertight seal. To prevent corrosion, the finished lysimeter was then coated with brown enamel paint.

The three loadcells were supported by leveling mounts made of stainless steel, shear-beam loadcells, and the corresponding wheels on which they were mounted. Each loadcell, with dimensions—16.5 × 9.0 × 4.5 cm and a weight of 300 g, Brand—generic. Model number MEP 08 (Dongguan South China Sea Electronic Co., Ltd., Dongguan 523132, China) had a 500 kg (1102.31 lb) capacity. Four holes of 25 × 25 single-ended sheer beam loadcells (Model CZL 642 Brand GREEN LABEL) were threaded inside the loadcells on both ends to support tanks (cultivation and drainage). The ground clearance was adjusted to ensure proper leveling of the cultivation tank and care was taken to ensure the even distribution of weight of cultivation tank on each loadcell. Isometric views of the loadcell mounting assemblies are shown in Figure 3. The dimension of different components of lysimeter as well as circuit diagram are shown in Figure 4 and Figure 5, respectively. The drain assembly consisted of a perforated sheet at the bottom of the tank, and a pipe outlet and collection tank were constructed 10 cm above the bottom of the wheels.

#### 2.3.2. Selection of Crop Type

The dimensions of the developed lysimeter were enough for planting six chrysanthemum plants in a row with plant-to-plant and row-to-row spacings taken as 15 cm and 30 cm, respectively, which were the same as the surrounding crop outside the lysimeter. There were two main reasons to adopt this popular greenhouse flower crop: (i) its higher planting density because of its large canopy size and (ii) its uniform shallow root depth in contrast to other small horticultural crops.

#### 2.3.3. Installation

The lysimeter was installed at the C.P.C.T., Indian Agricultural Research Institute, ICAR-IARI, New Delhi, inside the climate-controlled greenhouse. The soil type was sandy loam. Using a backhoe, hand tools, and shovels, installation was accomplished. The location of the lysimeter was marked, and the soil in the rectangular area was dug out to a depth of roughly 30 cm with a hand shovel. Then, the lysimeter was placed inside. The lysimeter was 60 cm deep. Soil inside the lysimeter was kept to a depth of 30 cm to maintain identical micro-environmental conditions for both the greenhouse and the lysimeter. Figure 6 represents the schematic diagram of Lysimeter installed in the middle of the bed, with proper management having a fixed plant-to-plant and row-to-row spacing. The soil surface inside the lysimeter is the same as the soil surface outside it. The triangular arrangement of wheels and loadcells at the time of fabrication are shown in Figure 7a and Figure 8 respectively. The canopy of the plants inside the lysimeter is also similar to that of the plants in the greenhouse outside the lysimeter (Figure 7b). The soil was removed from the greenhouse field in large blocks to conserve as much as existing soil structure, also to preserve the similar soil properties and the unbroken soil blocks were set aside to fill it inside the cultivation tank.

To accommodate the dimensions of the cultivation tank, the hole was enlarged. The loadcell assemblies were bolted to wheels below the lysimeter. Loadcell wires were routed to a common corner of the cultivation tank and brought up to one of the four sides of the cultivation tank, where it was connected to an Arduino assembly and datalogger. The lysimeter rested on the leveled ground. A datalogger was provided to record the evenly distributed load imposed on each loadcells. If the weights on each loadcell were not similar, the lysimeter was lifted out, the heights of the leveling mounts were adjusted and the loadcell weights were checked repeatedly to ensure its proper functioning.

#### 2.3.4. Structural Analysis

To ensure the proper functioning of the lysimeter, the total weight of the soil mass was retained on a perforated sheet constructed at the bottom of the cultivation tank. A gap was maintained between the perforated sheet and the bottom plain sheet of the cultivation tank for drainage of excess water from the soil mass. It also prevented element distortions and showed enhanced arrangements in the measurements of the weighing system. SolidWorks 2016 v.24 (Dassault Systèmes SolidWorks Corporation 175 Wyman Street Waltham, MA 02451, USA) was used to design the components of the lysimeter (Dassault Systèmes, S.A., Vélizy-Villacoublay, France) [32] whereas, SolidWorks Simulation components were used for the structural behavior. To examine the stress simulation and deformations in the side walls of the cultivation tank and perforated sheet as the base structure under different load conditions, a static analysis was performed. The lysimeter was installed in sandy loam soil with proper management. The features of this soil were extracted from the Technical Building Code (Ministry of Housing, 2006), having a bulk density of 15 KN/m^3^, an internal friction angle of 35° (ϕ) and an active earth pressure coefficient (K_a_) of 0.270. Soil was uniformly distributed on the lysimeter structures and the estimation of active lateral earth pressures (E_a_) due to soil load was done with the help of Rankine’s theory (Equation (1)):(1)$$E_{a}=K_{a}\times q$$
where q is load that may be present and K_a_ is coefficient of the active earth pressure (Equation (2)).
(2)$$K_{a}=\frac{1-\sin\phi }{1+\sin\phi }$$

For lateral earth pressure due to backfill on the sides of the cultivation tank, a uniform distribution was assumed whereas, a triangular distribution in the case of base/bottom structure of lysimeter for determining the variable q. From Equations (1) and (2), the maximum obtained value of uniformly distributed lateral earth pressure due to soil backfilled was calculated as 1650 N/m^2^ on each side of the cultivation tank and the load (self-weight including wheels weight + total Earth weight on perforated sheet at bottom) were 1027 and 6600 N/m^2^ (total of 7627 N/m^2^). The combinations of different load cases are shown in Table 2. SolidWorks Simulation was used for the analysis of the 3-dimensional models; it was essentially required to simplify the original models by which mesh size and computational resources would be allowed to optimized [37]. A finite element model was developed for the cultivation tank and the bottom structure, i.e., perforated sheet. “BEAM” type elements for the tubular profiles and “SHELL” type elements for the sheets profile were selected because of their lower thickness. Similarly, “SOLID”-type elements were used for the tubular profiles to model the finite elements of the bottom/base structure. The floor of the cultivation tank of the lysimeter acts as a compressible material deformed by the loads transmitted on the triangular arrangement of the wheelbase (Figure 9). There was a reaction of opposite pressure on the point of contact surface that was directly proportional to the displacement normal component. These assumptions were adopted from Finite Element Analysis Concepts: Via Solidworks.

#### 2.3.5. Acquisition of Data and Control System

The lysimeter weighing system consisted of three loadcells, Model number MEP 08 C, CZL-642 (Guangdong South China Sea Electronic Measuring Technology Company Ltd.), in accordance with MEP 08 class C regulations. The loadcells had a sensitivity of 2m V/V. The maximum holding load for each group of loadcells was about 500 kg, and its weighing precision was 20 g, which was precise enough for the correct measurements. The data obtained from the loadcells were recorded by axcluma micro-sd card reader module, Model no (BE-000011) and it supported micro sd card and micro sdhc card (high-speed card). The level conversion circuit board that can interface the level 5 V or 3.3 V power supply was 4.5 V to 5.5 V. The 3.3 V voltage regulator circuit board communication interface is a standard spi interface with 4 m^2^ screw positioning holes for easy installation of the control interface. A total of six pins (gnd, vcc, miso, mosi, sck, cs), gnd to ground, vcc is the power supply, miso, mosi, sck is the spi bus, cs is the chip select signal pin 3.3 V regulator circuit: Ldo regulator output 3.3 V as level converter chip, micro sd card supply level conversion circuit. Figure 10a represents the Arduino display box fixed at the side wall of cultivation tank and Figure 10b represents the circuit diagram of sd card reader module, a component of arduino assembly. Figure 10c represents Chrysanthemum grown on a lysimeter.

All components are controlled by arduino mega 2560, a microcontroller board based on atmega 2560. It has 54 digital I/O pins (of which 15 can be used as pwm outputs), 16 analog inputs, 4 uarts (hardware serial ports), a 16 MHz crystal oscillator, a usb connection, a power jack, an icsp header, and a reset button. Simply plug in a USB cord to a computer or power it with an AC-to-DC adapter or battery to get started. Most shields designed for the Arduino duemilanove or diecimila are compatible with Mega. The updated version of aruduino mega is mega 2560.

### 2.4. Operation

The datalogger was properly programmed and used to collect measurement records from the loadcells at every 1 h intervals. An excitation voltage was sent to the three loadcells with each loadcell read ten times, and the mean of ten readings was calculated. The total weight of the lysimeter and the total weight were stored in the backup storage module and the datalogger memory. The data were downloaded in a timely manner from the storage module and then input into a spreadsheet and finally imported to Excel for analysis and suitable graphical representation. There was essentially routine maintenance, which involved regular visits in the greenhouse and installed lysimeter area to examine the situation of vegetation growth within and around the lysimeter. Farm operations, such as tillage, fertigation and spraying, were also performed manually at regular intervals in order to match outside conditions inside the lysimeter. The estimation of hourly and daily crop evapotranspiration (ET_c_) were determined by subtracting the lysimeter weight from one reading from the next reading. Water evaporating from plant and soil surfaces, i.e., in the form of evaporation and transpiring through plant tissues, was the main reason for the weight loss of the cultivation tank. The change in weight, in kilograms (Kg), was converted to an equivalent depth of water, in millimeters (mm), by dividing the changed weight by the density of water (g/cm^3^) and the surface area of the inner tank (m^2^). Total lysimeter weight decreased continuously due to ET_c_ whereas, weight increased due to irrigation. The total change in water content on a daily basis was calculated by accumulating the hourly change in weight and converting it to equivalent water content. Daily changes were estimated by adding the 24-hourly changed weight starting from 00:00 on one day to 00:00 on the next following day and corresponded with daily microclimate data reported by the weather station located at the experimental greenhouse.

## 3. Results

This results section is divided into the following subheadings. It should reveal concise and precise explanations of the experimental results, their interpretation and, more importantly, experimental conclusions that can be drawn.

### 3.1. Calibration Process for Lysimeters

Before installation of the lysimeter, a calibration routine of the lysimeter’s loadcells was followed to confirm its proper functioning and accuracy. A combination of thirty-two known weights were placed one by one in the cultivation tank of the lysimeter, and corresponding output weights were recorded. The weight changes recorded by the loadcells were then examined and compared to the known weight changes, as shown in Figure 11. A regression equation has been developed to use this equation in the Arduino program for estimation of actual change in weight of lysimeter and results in accurate measurements from lysimeter. All loadcells accurately accounted for the change in weight for both increasing and decreasing cases. The description of the statistical analysis in the calibration process before installation of the lysimeter is shown in Table 3. The average error magnitude is shown by MAE and RMSE; however, they do not provide information on the average difference before and after calibration. The bias of the error is described by the MBE. However, its importance depends on the size of the data being examined. A negative MBE occurs when predictions are smaller in value than observations.

### 3.2. Structural Analysis of Lysimeter

According to the structural analysis, the maximum possible deformations that each structure could have undergone under the various load cases were not greater than their parting distance (Table 3). The highest deformation measured for the cultivation tank was 0.6137 mm, and the Von Mises equivalent stress was 12.77 MPa (Table 4, Figure 11). The highest vertical displacement for the perforated sheet, which makes up the bottom structure, was 7.1 mm, and the Von Mises equivalent stress was 66.7 MPa (Table 4). For the cultivation tank and bottom perforated sheet, the safety factors were 19.2 and 3.7, respectively. The bottom of the lysimeter showed minimal overall displacements/deformations for the kind of soil and the various loading cases taken into consideration in this study. In any case, the elastic limit of the mild steel used in the lysimeter was not exceeded by the Von Mises equivalent stress of the designed bottom (Figure 12).

### 3.3. Operation Results

Table 5 reveals the outcomes of the water balance, as well as the weight differential.

The factors involved in the irrigation process in the plant–water–soil system were determined using variations in weight. The water balance showed precise measurements of water losses throughout the entire crop season using a developed lysimeter (Table 5). The cultivation tank’s weight increased due to irrigation, but it then fell as the crop began to take water, as expected (Figure 13). When the water first began to drain through the soil, the weight of the cultivation tank was reduced rapidly, followed by a gentler decline due to the water consumption by the crops. There was no scenario of precipitation events inside the greenhouse to increase the water content beyond the field capacity. Weight fluctuations were identified in the cultivation tank of the installed lysimeter. The results revealed that there was a slight increase at night, which may be due to condensation and diminished during the day. The hourly recorded weights of cultivation tanks inside greenhouses at different plant growth stages are shown in Figure 13.

### 3.4. Crop Evapotranspiration (ET_c_)

The measured crop evapotranspiration (ET_c_) obtained from the lysimeter and the reference evapotranspiration (ET_o_) acquired from a pan evaporimeter installed inside the greenhouse for the chrysanthemum crop is described in Figure 13. The values of ET_c_ and ET_o_ varied from a low of 1.70 and 1.84 mm/day during the vegetative stage to a high of 10.19 and 13.52 mm/day flowering stage. The average values of the ET_c_ were 1.19, 4.96 and 3.17 mm/day in the initial stage, mid-season stage, and late season stages, respectively. Similarly, the values of ET_o_ were 1.29 mm/day in the initial stage, 6.41 mm/day in the mid-season stage and 4.89 mm/day in the late season stages. Overall, the values of ET_c_ were somewhat close to the ET_o_ values. However, the values of ET_o_ were overestimated by 0.15 mm in 2019–2020 and underestimated by 1.28 mm in 2020–2021 compared to the values of ET_c_ and are represented in Figure 14.

### 3.5. Crop Coefficient of Chrysanthemum

The crop coefficient (K_c_)values of the chrysanthemum crop estimated from the lysimeter installed inside the greenhouse for crop seasons (2019–2020 and 2020–2021) are presented in Figure 14. The generated values from the lysimeter were K_c_ ini 0.43 and 0.38, K_c_ mid-1.27 and 1.25 and K_c_ end-0.67 and 0.59 for the years 2019–2020 and 2020–2021, respectively. The results revealed that the values of K_c_ were comparatively lower at the initial stage of crop growth, primarily due to the fact that evapotranspiration was mainly affected by evaporation, as the crop canopy had very small ground coverage. On the other hand, the values of K_c_ were highest during the mid-season stage due to significantly high evapotranspiration. The K_c_ values decreased progressively at the late-season stage due to the full maturity and aging of the leaves simultaneously. The results were then compared to the FAO (Food and Agriculture Organization) tabulated results for K_c_, the obtained K_c_ values from the developed lysimeter followed a similar trend as the FAO recommended crop coefficient values for crops of the same family, although the crop coefficient of chrysanthemum crop was not discussed in the FAO-56 manual [1].

## 4. Discussion

The developed small weighing lysimeters, such as the Smart Field Lysimeter [33] and the Ready-To-Go lysimeter with cylindrical shapes, were used for field-based water management studies [32]. The Smart Field Lysimeter (SFL) had a 300 mm diameter and different depths of 300, 600 and 900 mm of the cultivation tank. The Ready-To-Go field lysimeter had models with 300- or 800-mm diameter and different depths of 300, 600 and 900 mm of the cultivation tank. The above-developed weighing type lysimeters of cylindrical shape and their constructional dimensions were not suitably adopted for horticultural crops for accurate measurement of ET_c_ related data [33]. In the case of traditional weighing lysimeters, the loss or gain of water is estimated by the change in weight obtained by weighing the whole container in which the soil is placed. To avoid these complicated tasks, Mishra et al. (2011) proposed a small lysimeter of low cost with a capacity of only 20 kg. It was constructed for a glasshouse and showed little accuracy in the measurement of ET_c_. The lysimeter installed for greenhouse sugarcane at pre-sprouted plantlets of Libardi et al. [38], Evapotranspiration and crop coefficient (K_c_) of pre-sprouted sugarcane plantlets for greenhouse irrigation management and the weighing lysimeter with triangular arrangement for potted plants of Ruíz-Peñalver et al. [39], were developed solely for research purposes and not upscaled for commercial use by farmers. Based on the above constraints, a weighing type portable automatic lysimeter was designed, developed and evaluated for shallow rooted greenhouse chrysanthemum flower crop with dimensions (100 × 100 cm and 80 cm depth) and a relatively high load carrying capacity of 1500 kg. The portable weighing-type lysimeter described in this paper was used from September to February during the 2019–2020 and 2020–2021 growing seasons to grow greenhouse chrysanthemum crops at the Center for Protected Cultivation Technology (C.P.C.T, ICAR—IARI, New Delhi). The system performed satisfactorily during both years and measured data were collected for further determination of crop water requirements for greenhouse chrysanthemum. The ability to move the lysimeter to the desired place randomly within an experimental plot was an advantage. This helped overcome the problems of spatial variability associated with field studies. The effect of wind was negligible inside the greenhouse and it did not affect the ET_c_ measurement. The lysimeter was placed very close to the ground on three loadcells below it to stabilize it during the measurement period. The developed lysimeter with portable attributes assembly for easy and safe handling provided accurate readings related to ET_c_ for the planting framework. To monitor temperature variation, temperature sensors were also installed inside the lysimeter and in the field at depths of 5, 10 and 15 cm below the soil surface to assess the representativeness of the lysimeters. The temperature sensors were placed inside the lysimeter 5 cm from the edge and in the middle of the lysimeter. There was no apparent temperature variation between the lysimeter and the field at 10 cm and 15 cm depths. However, as the surrounding temperature rose, the temperature at the 5 cm depth changed. The lysimeter’s edge temperature was somewhat higher, whereas the center temperature closely matched the field temperature. The temperature difference between the center and the edge increased as the ambient temperature rose. It has the potential for commercialization and large-scale adoption by greenhouse growers and can simultaneously be used for research purposes.

## 5. Conclusions

Overall, the developed lysimeter performed satisfactorily and the weighing system provided reliable data that can be used to determine crop water requirements. The purpose of this work is to develop a convenient lysimeter and to improve the limitations of traditional lysimeters. The following key findings have been drawn from this work.

The developed weighing type lysimeter is cheap, lightweight, portable and has automation features.It can be adopted by local greenhouse farmers to grow shallow-rooted crops for efficient water and nutrient management.The developed lysimeter is sensitive to sudden load variations. Uncertainty of the weight measurement due to sudden abrupt external forces might disturb the readings.It has generated crop coefficient (K_c_) for greenhouse chrysanthemum crops, which was missing in the existing literature.The development of a temperature gradient at the lysimeter sides and edge is a cause for concern. However, since the growth of flowers in this region is done during the winter months in a greenhouse with relatively uniform temperatures under structure, it has no significant impact on the results of the crop grown in this lysimeter.

## Figures and Tables

**Figure 1 sensors-22-06239-f001:**
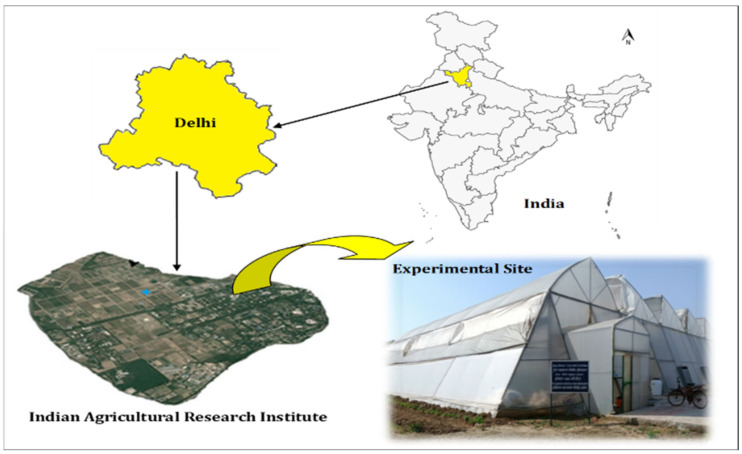
Location of ICAR—Indian Agricultural Research Institute, New Delhi.

**Figure 2 sensors-22-06239-f002:**
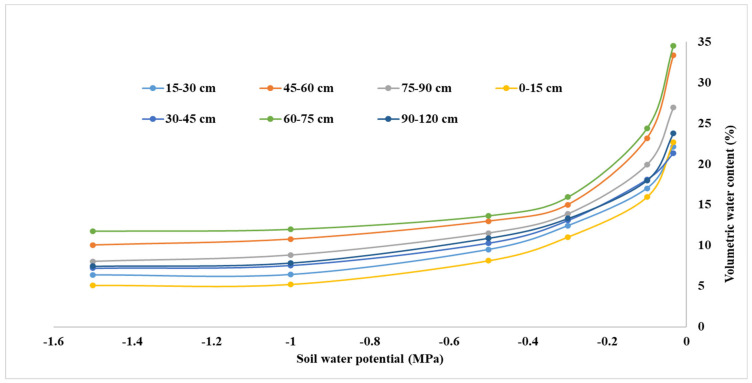
Soil moisture characteristics curve of the experimental site, showing the curves for seven different depths.

**Figure 3 sensors-22-06239-f003:**
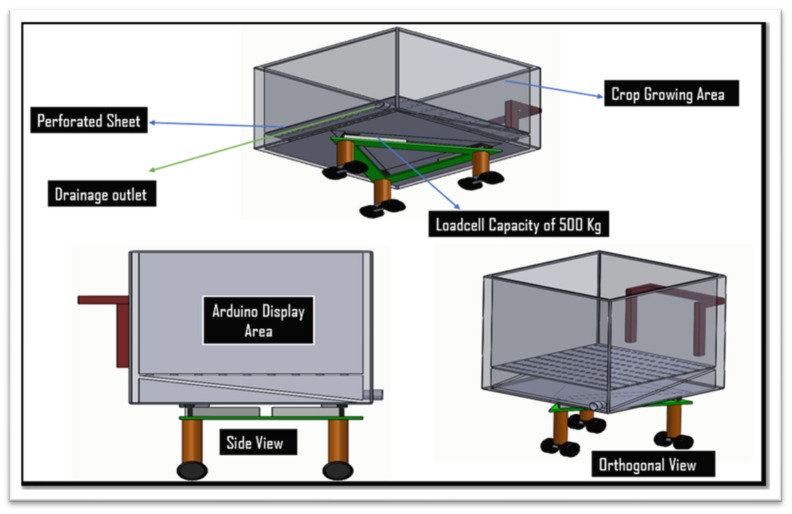
Design drawing of different components of a lysimeter suitable for chrysanthemum crops.

**Figure 4 sensors-22-06239-f004:**
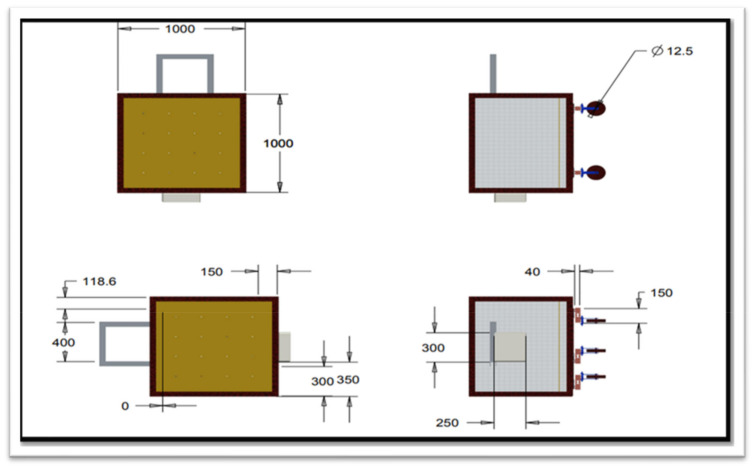
Dimensions of the various lysimeter components, selected for a chrysanthemum plantation, were 1000 mm × 1000 mm × 600 mm.

**Figure 5 sensors-22-06239-f005:**
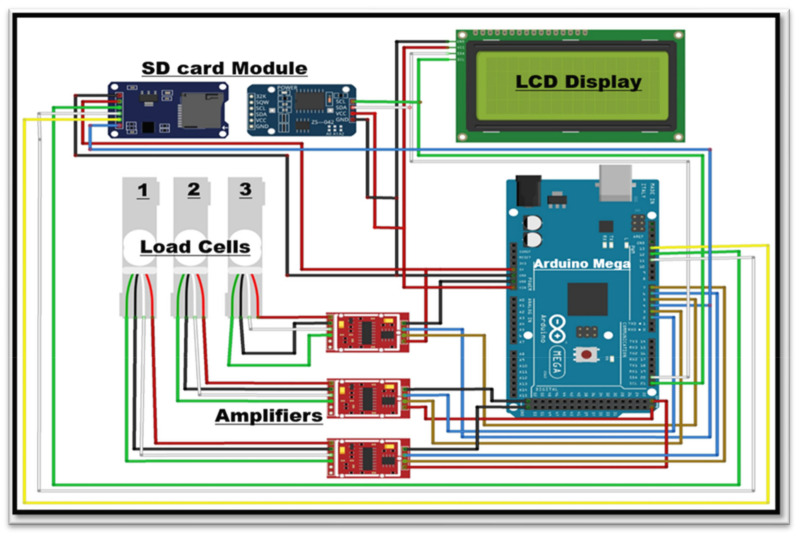
Circuit diagram of different components of the lysimeter.

**Figure 6 sensors-22-06239-f006:**
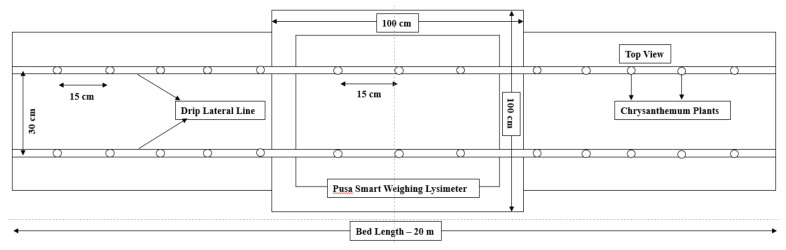
Schematic diagram of Lysimeter installed in the middle of the bed with proper management having a fixed plant-to-plant and row-to-row spacing.

**Figure 7 sensors-22-06239-f007:**
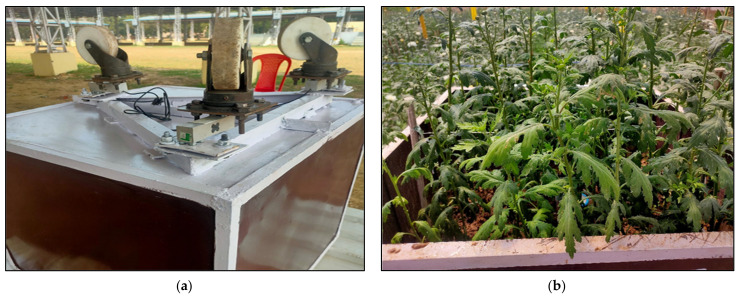
(**a**) Triangular arrangement of wheels at the bottom of the lysimeter; (**b**) the lysimeter installed in the greenhouse and crops are grown.

**Figure 8 sensors-22-06239-f008:**
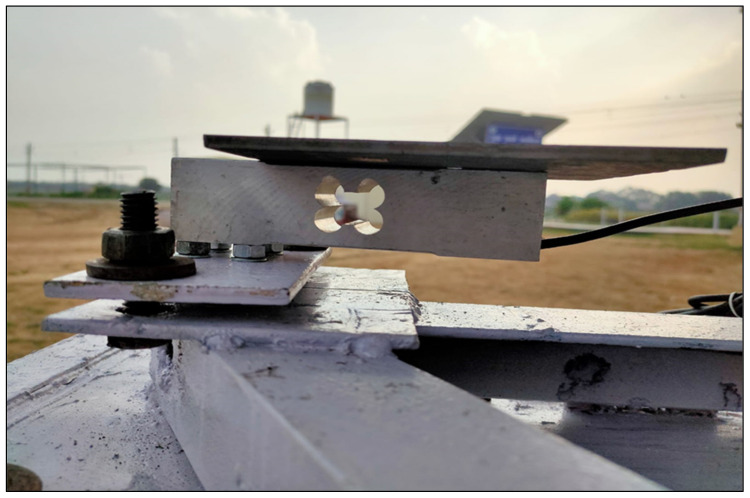
Arrangement of loadcell assembly at bottom structure of the lysimeter.

**Figure 9 sensors-22-06239-f009:**
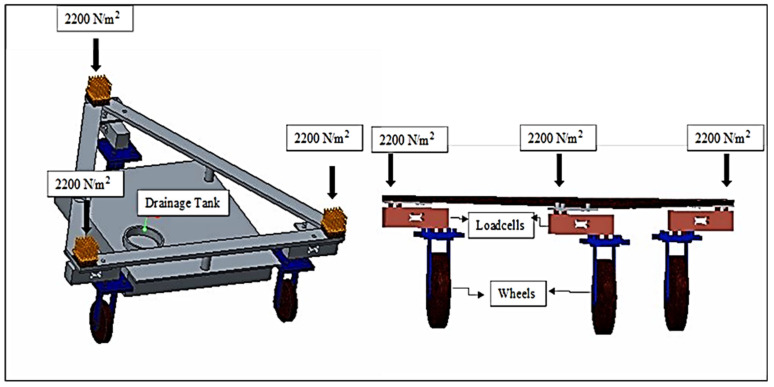
Triangular load distribution acts on loadcells at the bottom structure of the lysimeter.

**Figure 10 sensors-22-06239-f010:**
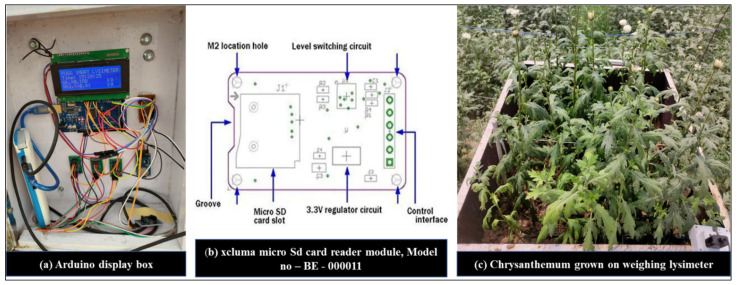
(**a**) Arduino display box, (**b**) Sd card reader module, and (**c**) Chrysanthemum grown on a lysimeter.

**Figure 11 sensors-22-06239-f011:**
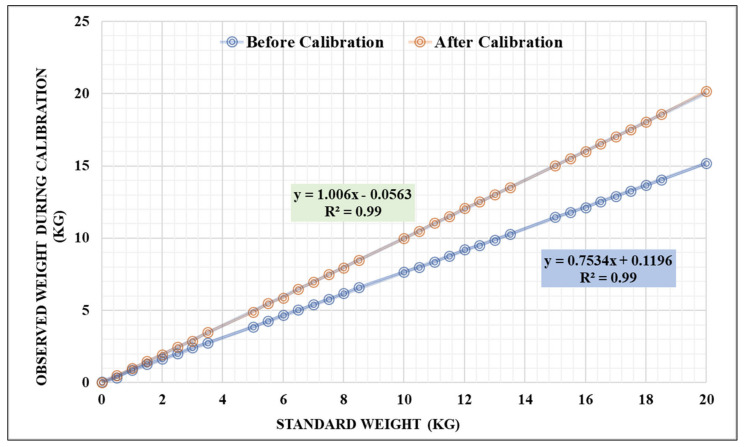
Calibration results for the lysimeter before plantation of the chrysanthemum crop.

**Figure 12 sensors-22-06239-f012:**
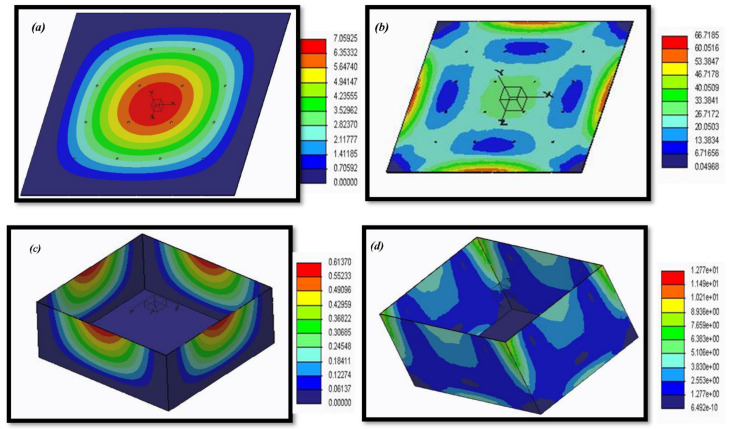
Three-dimensional view of the results obtained in load combination: (**a**) and (**c**) represent resulting displacement (mm), whereas (**b**) and (**d**) represent Von Mises equivalent stress (MPa) for the bottom perforated sheet and walls of the cultivation tank of the lysimeter, respectively.

**Figure 13 sensors-22-06239-f013:**
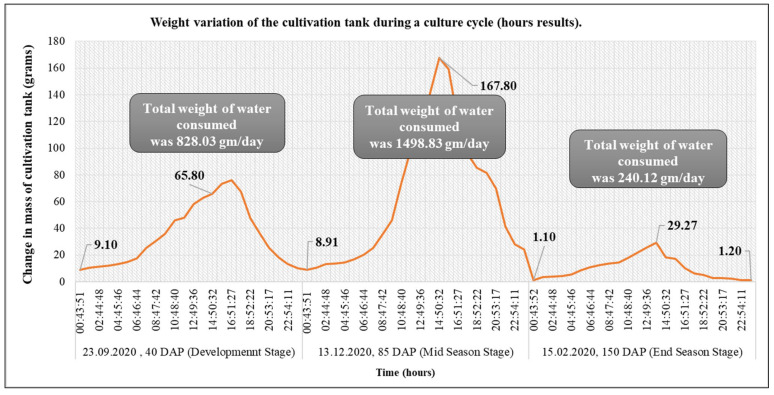
Hourly chrysanthemum lysimeter data showing total weight and cumulative change in water content for a three-day period during different plant growth stages.

**Figure 14 sensors-22-06239-f014:**
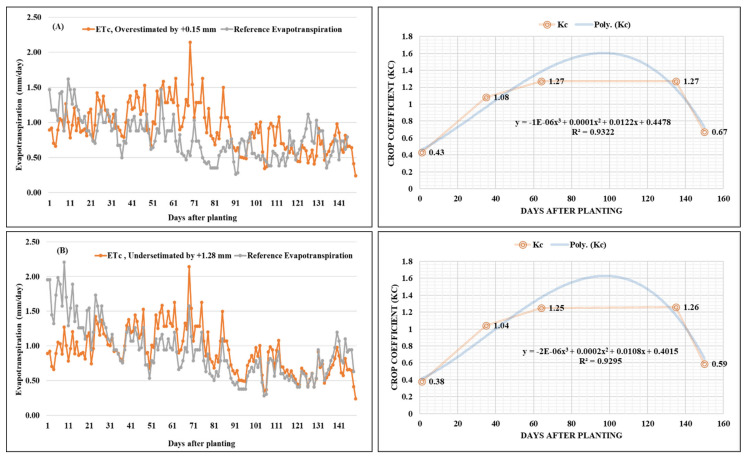
Crop coefficient, K_c_, of the chrysanthemum (variety Zembla) crop at (**A**) 2019–2020 and (**B**) 2020–2021.

**Table 1 sensors-22-06239-t001:** Planting and harvest dates with seasonal weather data from 2019–2021.

Properties	2019–2020	2020–2021
Planting Date	20-09-2019	10-10-2020
Last Harvesting Date	16-02-2020	07-03-2021
Mean Air Temperature (°C)	20.8	19.8
Mean Relative Humidity (%)	72.4	73.0
Mean Solar Short-wave radiation (Watt/m^2^)	275.5	278.9
Total Rainfall (mm)	Zero	Zero
Lysimeter ET_c_ (mm)	152.6	147.8
Class-A Pan ET_o_ (mm)	153.9	147.9

**Table 2 sensors-22-06239-t002:** Different load cases were considered for the cultivation tank and the base structure.

	Load Case	Value of the Load (N/m^2^)	Load Distribution
Cultivation tank	Self—weight of lysimeter excluding wheels weight	784.8	Uniform
	Lateral earth Pressure	1650	
Base structure	Self—weight including wheels weight	1027	Triangular
	Total Earth weight on perforated sheet at bottom	6600	

**Table 3 sensors-22-06239-t003:** Descriptions of statistical analysis in the calibration process of the lysimeter.

Statistical Indices	D	RMSE	RMAE	MBE	MSE	MAE	RE
Before Calibration	0.99	2.02	0.20	−1.33	4.18	1.33	0.10
After Calibration	1.00	0.04	0.02	0.00	0.00	0.02	0.00

Note: D—Index of Agreement. RMSE—Root Mean Squared Error, RMAE—Square Root of the Mean Absolute Error, MBE—Mean Bias Error, MSE—Mean Squared Error, MAE—Mean Absolute Error, RE—Relative Error.

**Table 4 sensors-22-06239-t004:** Results of the analysis for the sides of the cultivation tank, its main structure and the base structure of the lysimeter.

Load Case	VMS (MPa)	URES: R.D (mm)	F.S
Lateral earth pressure on cultivation tank	12.7	0.6	19.2
Total earth weight on perforated sheet at bottom	66.7	7.1	3.7

Note: VMS—Von Mises Equivalent Stress (used to predict yield or fracture of materials when subjected to a complex loading condition, mostly used for ductile materials), R.D—Resulting Displacement, F.S—Factor of safety.

**Table 5 sensors-22-06239-t005:** Results for the water balance generated from the smart weighing lysimeter.

	Components	2019–2020			2020–2021		
		Water Film (mm)	Mass (g)	Difference In-O (g, mm)	Water Film (mm)	Mass (g)	Difference In-O (g, mm)
**In**	**R**	0	0	+8976 +8.9	0	0	+7410 +7.4
	**I**	153.9	153,961		147.8	147,967	
	**C**	7.7	7698		7.4	7391	
**O**	**ET_c_**	152.6	152,683		147.9	147,948	
	**D**	0	0		0	0	

Note: In—Inputs, O—Outputs, R—Rainfall, I—Irrigation, C—Condensation, ET_c_—Crop evapotranspiration, D—Drainage.

## Data Availability

Not applicable.

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
