# Peer review of "Development of Smart Weighing Lysimeter for Measuring Evapotranspiration and Developing Crop Coefficient for Greenhouse Chrysanthemum"

_sensors, 2022, doi:10.3390/s22166239_

Round 1
Reviewer 1 Report
The objective of this manuscript submitted in Sensors is to design, construction, and installation of a smart weighing lysimeter to measure the crop evapotranspiration for shallow–rooted Chrysanthemum flower crop. The authors described well the purpose of study and the actual problem.
Materials and Methods
Line 136, Correct the Units of Electrical conductivity ds/ to dS/m
Figure 2. It is suggested to present soil water potential in mega pascal rather than kpascal. smaller Units with large values may cause confusion.
Description of statistical analysis of the data (calibration, operation etc.) is missing
Resulst
Fig 13. Font size is small and difficult to read text as well as data numerals.
Fig 14 Increase the font size to increase the readability
Discussion
This section is too short and various key points of measurement of Evapotranspiration and crop coefficient in newly designed lysimeter with various modifications are missing. Some points of Discussion section are presented in Conclusion section, and Conclusion section is too long. It is suggested to revise Discussion and Conclusion section. Conclusion section should not be more than 4-5 lines.
Reviewer 2 Report
Manuscript ID: sensors-1830290
Article Title: Development of Smart Weighing Lysimeter for Measuring Evapotranspiration and Developing Crop Coefficient for Greenhouse Chrysanthemum
Comments
This study focused on Development of Smart Weighing Lysimeter for Measuring Evapotranspiration and Developing Crop Coefficient for Greenhouse Chrysanthemum. The article has scope to publish in the journal but not in the present form. Introduction needs refinements. Methodology is poorly written. Many sentences are unclear in the manuscript. Tables need to be re-formulated. Results need lot of improvements. I would recommend for major corrections.
Comment 1: The abstract doesn’t show the accurate content and the main findings of the study area. Please add the main findings of the research work and check the performance of the developed lysimeter for another crops.
Comments 2: I recommend the authors to write in the Introduction more explicitly based on existing literature what is missing in previous studies, what is the added value of this new study.
Comment 3: Please proofread the article carefully; there are many linguistic errors in the manuscript.
Comment 4: In general, the manuscript is written like a report summary.
Comment 5: The figures 6, 9, 10 and 11 have poor quality. The legibility of the figures are not appropriate please update with 300 dpi or more.
Comment 6: The methodology section is very unclear and needs updates.
Comments 7: Please add more discussion material to the “Discussion” section. What were perhaps different results from other studies and why?
Comments 8: The conclusion should be specific. It is recommended to just highlight the key findings of the work.
The article has scope to publish in the journal but not in the present form. In general, the manuscript is written like a report summary. The introduction needs refinements. The methodology is poorly written. Many sentences are unclear in the manuscript. Tables need to be re-formulated. Results need a lot of improvements. I would recommend major corrections.
Round 2
Reviewer 2 Report
The article has the scope to publish in the journal in the present form. English language and style are fine/minor spell check is required. I would recommend accepting it in the present form.